# Retrospective Analysis of Fever in Pediatric Age: Our Experience over the Last 5 Years

**DOI:** 10.3390/children11050539

**Published:** 2024-04-30

**Authors:** Mariella Valenzise, Federica D’Amico, Giulia La Barbera, Carlo Maria Cassone, Silvia Patafi, Fortunato Lombardo, Tommaso Aversa, Malgorzata Gabriela Wasniewska, Giuseppina Salzano, Carmela Morace

**Affiliations:** 1Department of Human Pathology of the Adulthood and Childhood, University of Messina, 98121 Messina, Italy; federica.damico1@studenti.unime.it (F.D.); giulia.labarbera@asst-fbf-sacco.it (G.L.B.); cssclm94a01h224y@studenti.unime.it (C.M.C.); ptfslv95l46f112i@studenti.unime.it (S.P.); fortunato.lombardo@unime.it (F.L.); tommaso.aversa@unime.it (T.A.); malgorzata.wasniewska@unime.it (M.G.W.); gsalzano@unime.it (G.S.); 2Department of Clinical and Sperimental Medicine, University of Messina, 98121 Messina, Italy; carmela.morace@unime.it

**Keywords:** fever, epidemiology, pediatric

## Abstract

Background: Fever is one of the most frequent symptoms highlighted during medical assistance. Due to this great impact, our study has the purpose of analyzing the demographic and laboratory characteristics of patients hospitalized in our center and identifying predictive markers to make the differential diagnosis between infectious and non-infectious fever. Methods: Our population included 220 children, collected from January 2017 to August 2022, hospitalized for continuous fever (4 days or more in duration with at least one temperature peak ≥37.5 °C) and excluded cases of discharge against medical advice and/or transfer to other operating units. Demographic (mean age at the time of admission, frequency of hospitalization, and mean days of hospitalization), laboratory, and instrumental variables were analyzed in order to find correlation with fever etiology. Results: Older age at the time of hospitalization, family history of periodic fever, fever lasting more than 8 days, and longer hospitalization are strongly associated with non-infectious fever, together with anemia, high platelet count, high CRP and ferritin, and hyponatremia at the time of admission. Paracetamol is the preferred antipyretic treatment. Echocardiogram has shown anomalies in patients with infectious fever, while ECG anomalies were detected in non-infectious fever. Conclusions: Our data underline the importance of predictive markers, such as clinical and laboratory parameters, to differentiate infectious from non-infectious fevers, but further studies are necessary.

## 1. Introduction

Fever is one of the most important defense mechanisms of the human body, and it is defined by a rectal temperature of ≥38 °C (100.4 °F) or axillary temperatures of ≥37.5 °C (99.5 °F) [1]. It is one of the most common reasons for medical assistance [2].

It is important to distinguish fever from hyperthermia and the use of different drugs. In the first case, the exposure to high environmental temperature is followed by the elevation of body temperature without a compensation of thermoregulation mechanisms [3]. In the second case of drugs, such as antiepileptics, furosemide and levothyroxine, they are linked to impaired thermoregulation [4]. This difference is related to an opposed pathophysiological mechanism; in fever, body thermoregulation is set to higher threshold values, and in hyperthermia, there is an increase of the central body temperature. This occurs through various mechanisms such as the inhibition of heat dispersion, excessive heat production, and/or hypothalamic dysregulation [5].

Thermoregulation is the result of a balance between heat production and dispersion, mediated by the interaction between different structures located in the hypothalamus and in the preoptic region, in the endocrine glands (thyroid, adrenals), or in peripheral (sweat glands, mucous membranes, adipose tissue, and vascularization of skin surfaces) [6]. During fever, exogenous and endogenous pyrogens reset the thermoregulatory pathway in these two different ways: the humoral and the neural pathways. In the first one, there are pathogen-associated molecular patterns (PAMPS) or pyrogenic cytokines that bind to toll-like receptors 4 (TLR-4) located on the fenestrated capillaries in the blood–brain barrier. This is followed by the release of prostaglandin E2 (PGE2), which activates thermal neurons in the anterior hypothalamus to a higher thermal balance point [7,8]. In addition, the humoral pathway is related to the circulation of pyrogenic cytokines [9]. Conversely, the neural pathway is characterized by cold-sensitive cutaneous nerves and vagus stimulations by PGE2, cytokines, and norepinephrine. These pathways transmit fever signals to the nucleus of the tractus solitarius and the preoptic and hypothalamic areas [9].

According to the duration, fevers can be distinguished into acute (<7 days in duration), subacute (not more than 2 weeks in duration), and chronic (>2 weeks in duration) [1]. Based on the elevation of rectal temperature, fever can be differentiated into low (between 38.1° and 39 °C), moderate low (between 39.1° and 40 °C), high -grade low (between 40.1° and 41.1 °C), and hyperpyrexia (>41.1 °C) [1]. Regarding the trend of the thermal curve, fever can be defined as continuous, remitting (daily oscillations of 2–3 °C without defervescence), intermittent (alternation of fever-apyrexia), and undulating (prolonged fever alternating with periods of low-grade fever, without ever completing defervescence) [1].

In the pediatric age, fever represents one of the most frequent symptoms and the second reason for visits to the emergency room [10], probably due to the phenomenon of fever phobia. This is the exaggerated and irrational fear of fever of parents, caregivers, and healthcare personnel [11], probably related to the risk of convulsion, dehydration, cerebral damage, and coma [12]. The international literature shows various diagnostic protocols to distinguish low-risk situations from the high possibility of severe bacterial infection. However, none of them were demonstrated to be superior to the others [13,14]. The most important risk factors for severe bacterial infection are an age < 3 months of life, comorbidities, and immunodeficiencies [15]. Clinical observation remains the key point in cases of children with fevers [16].

According to the great impact of fever, the aims of this study are the following: to compare our epidemiological data with that of international literature, to establish predictive markers for infectious and non-infectious fever, and, consequently, to decide which patients need a pinpointed analysis and hospitalization.

## 2. Materials and Methods

A retrospective analysis was conducted, between January 2017 and August 2022, on 220 pediatric patients hospitalized at our clinic. Hospitalization criteria were as follows: unresponsive fever, necessity of intravenous hydration and/or antibiotic, and leukocytosis and/or protein C-reactive (CRP) ≥ 0.5 mg/dL.

The eligibility criteria were as follows: age ≤ 16 years with the presence of continuous fever (4 days or more in duration) and with at least one temperature peak ≥37.5 °C during hospitalization. Patients discharged against medical advice and/or transferred to other operating units (e.g., pediatric surgery) were excluded from the study. Patient data were collected from their electronic medical records.

For each patient, we considered demographic characteristics (age at the time of admission, sex, ethnicity), family history for periodic fever, days of fever, duration of hospitalization, and therapies during hospitalization (paracetamol and/or ibuprofen, intravenous rehydration, antibiotic therapy, systemic steroid therapy, and intravenous immunoglobulin infusion). The laboratory analysis was performed in two distinctive moments, the time of hospitalization (T1) and the time of discharge (T2). From a laboratory perspective, the analysis was conducted on blood count (white blood cells (WBC), percentage of neutrophils, lymphocytes, monocytes and eosinophils, hemoglobin (Hb), platelet (PLT)), CRP, erythro-sedimentation rate (ESR), procalcitonin, ferritin, serum iron, fibrinogen, triglycerides, D-dimer, transaminases (GPT, GOT and GGT), sodium, lipase, amylase, lactic dehydrogenase (LDH), T troponin, pro-B-type natriuretic peptide (pro-BNP), and myoglobin. Furthermore, we evaluated the following diagnostic tests and their results: urinalysis; urine culture; blood culture; measurement of fecal calprotectin; culture test on throat-tonsillar swab; measurement of anti-nuclear antibodies (ANA), anti-extractable nuclear antigens (ENA), anti-native DNA (nDNA), anti-neutrophil cytoplasmic antibodies (ANCA), anti-cardiolipin antibodies; viral serologies (EBV, CMV, parvovirus, adenovirus, rotavirus, HAV, HBV, HCV and SARS-CoV-2); serologies for mycoplasma pneumoniae and chlamydia pneumoniae; and instrumental tests such as chest and abdomen X-ray, electrocardiogram (ECG), echocardiogram-color Doppler, and abdomen ultrasound.

At the end, patients were divided, according to the etiology of the fever, into the 4 following groups: infectious (I), rheumatological (R), hematological (H), and other (O). Due to the small number of some subgroups, two large sets were considered for statistical purposes, infectious (I) and non-infectious (NI). In patients with a suspicion of infective fever, laboratory and radiological tests were performed in order to find the infectious focus. Children with a history of fever without an origin of infection and weight loss and who were unresponsive to antibiotics were referred for a possible diagnosis of fever of non-infectious origin. In patients where a non-infectious fever was suspected, other tests were performed such as autoimmunity (ANA, nDNA, ANCA, ENA, anti-cardiolipid antibodies), peripheral blood smear, and lymphocyte typing, according to clinical suspicion (Figure 1).

## 3. Statistical Analysis

The statistical analysis was carried out using the statistical program SPSS (Statistical Package for Social Sciences), version 22.0 for Windows. Continuous variables were expressed as median (range) or mean ± standard deviation. Categorical variables were expressed as numbers and percentages. The Kolmogorov–Smirnov test was used to verify the normal distribution of the variables. The student’s *t*-test was used to compare the difference in distribution of continuous variables between the two study groups (infectious and non-infectious etiology). The chi-square test was used to compare the difference in distribution of the categorical variables in the two groups. It was considered statistically significant a *p*-value < 0.05.

## 4. Results

A total of 220 hospitalized patients were analyzed with a diagnosis of fever lasting 4 or more days. At the time of admission, a mean age of 4.73 years ± 4.3 was estimated. The mean days of hospitalizations were 9.15 (range 3–44 days) and 7.15 mean days of fever (range 4–30 days).

The demographic characteristics are schematically reported in Table 1. The total males were 128 (58.2%) and total females were 92 (41.8%). The M:F ratio was in favor of the male sex (1.4). The population studied was almost entirely made up of Caucasian subjects of Italian origin 209 (95%). Only 11 patients (5%) were foreigners. A positive family history of periodic fever (e.g., familial Mediterranean fever, PFAPA) was found in 8 (3.6%) patients. The distribution of the hospitalizations during the time of observation was mostly homogeneous.

Regarding the antipyretic therapy during hospitalization, 124 patients (56%) were treated with paracetamol, 21 patients (10%) with ibuprofen, 36 patients (16%) with a combination of paracetamol and ibuprofen, and 36 patients (16%) with paracetamol and subsequently with ibuprofen. Thirty-nine patients (18%) had not taken any antipyretic therapy.

Of the patients, 175 (79.5%) underwent oral or intravenous antibiotic therapy, which was started before or during hospitalization, 155 (70.5%) were rehydrated intravenously with a pediatric balanced physiological or electrolyte solution, 57 (25.9%) had taken systemic steroid therapy (orally and intravenously), and 30 (13.6%) had undergone intravenous immunoglobulin infusion.

The laboratory evaluation was performed during the hospitalization (T1) and upon discharge (T2) (Table 2). The blood-count examination revealed neutrophilia related to T1, with an average value of total WBCs in reduction at T2. In addition, a tendency to normalization of the leukocyte formula by age was observed. PLTs were normal for age at T1, with an increasing trend at discharge. Hb values were at the lower limits for age at T1 and with normal values at T2. At the time of admission, CRP was increased by approximately 10 times the normal value (mean value 5.13 mg/dL ± 5.8), instead of T2 which has shown a clear reduction (mean value 1.47 mg/dL ± 3). The mean value for ESR was 59.2 ± 33.8 at T1 with a decreased tendency in T2 (49.1 ± 33.3). D-dimer was above the normal value at both T1 and T2, with a decreasing trend at discharge. The other parameters analyzed did not show elevated values or a particular trend.

According to the clinical-anamnestic history, other biohumoral and instrumental tests were performed. One hundred and fifty (68%) patients underwent urinalysis, with a pathological result in 29 patients (19%). Among these, 43 (28.7%) performed a urine culture, and 6 (2.7%) had a pathological result. Eighty-five patients (38.6%) underwent a blood culture, performed during a fever peak, which was found to be pathological in nine patients (4.1%). Fecal calprotectin was measured in 6 children (2.7%), with a pathological value in 2 of them (33%). In 72 (32.7%) patients, culture examination was performed on a throat-tonsillar swab, with pathological results in 12 cases (5.5%).

The autoimmunity assay was performed in 18 patients. ANA positivity was found in 4 patients (1.8%) and anti-cardiolipin antibodies in one. ENA, nDNA, and ANCA were measured, and no pathological results were found.

EBV serology was performed in 116 patients (52.7%); 19 (16.4%) had a previous infection and 13 (11%) had a current infection. Serology for CMV was performed in 97 patients (44.1%). Twelve (13%) had a previous infection and five (5.5%) had a current infection. Anti-parvovirus antibodies were measured in 32 (14.5%) children, with a result of a current infection in 3 of them (1.4%) and a previous infection in 5 (2.3%). Adenovirus infection was investigated in 88 patients (40%), and a positive result was found in 6 (2.7%). A previous infection was found in 20 children (9.1%). The rotavirus test was performed in 29 (13.2%) patients with a negative result in all of them. The serologies for hepatitis viruses (HAV, HBV, and HCV) were performed in 13 (6%) patients (all negative for current infection). On the contrary, serology for mycoplasma pneumoniae was performed in 57 patients (25.9%). Six patients (10.5%) had a current infection and one (1.7%) had a probable previous infection. Serology for chlamydia pneumoniae was carried out in 51 patients (23.2%); 1 (1.9%) patient had a current infection, and 1 (1.95) had a previous infection. Serology for SARS-CoV-2 was performed since March 2020 in 82 (37.3%) patients; 13 cases (5.9%) had a result compatible with previous infection and/or vaccination.

Regarding the instrumental evaluation, 78 (35.45%) patients underwent a chest X-ray, which was pathological in 37 cases (16.8%). Fourteen patients (6.36%) underwent an abdominal X-ray that was pathological in four cases (1.8%). In addition, 73 patients performed abdominal ultrasound, which was pathological in 44 patients (20%). Among all patients, 105 (47.72%) had performed cardiological evaluation with a color Doppler echocardiogram, which was pathological in 43 patients (19.5%), and ECG in 64 patients, which was pathological in 19 of them (8.6%).

Finally, our population was divided into four subgroups, according to the following etiology of fever: infectious (I) forms in 158 patients (71.8%), rheumatological (R) in 49 (22.3%), hematological (H) in 4 (1.8%), and other causes (O) in 8 patients (3.6%) (such as inflammatory bowel disease). One patient presented a mixed form of fever (infectious and rheumatological). A deeper analysis is highlighted in Table 3.

For the greater prevalence of the infectious etiology and the small number of the remaining subgroups, we divided the population into these two main subgroups: infectious (I) and non-infectious forms (NI). All results were analyzed according to this classification. Regarding the demographic characteristics, a statistically significant result was found between NI etiology and the following items: age at hospitalization (*p* = 0.03), family history of periodic fever (*p* = 0.003), duration of fever (*p* = 0.023), and days of hospitalization (*p* = 0.000). The results are illustrated in the Table 4.

Regarding the laboratory data performed at T1 and T2, we have found a correlation between lymphocytosis and type I etiology both at the time of hospitalization and at discharge (*p*-value 0.021 and 0.004, respectively), instead of NI forms which correlated with a higher eosinophil value at T1 (*p*-value = 0.012). The blood count tests performed at discharge confirmed the correlation between lymphocytosis and type I forms and neutrophilia associated with NI forms (*p*-value = 0.004 and 0.007, respectively). Hb values were below normal limits at the time of admission (*p*-value = 0.011) and at discharge (*p*-value = 0.023), and they were correlated with NI forms. The same trend was highlighted by high PLT count at T2 (*p*-value = 0.000). CRP performed at T1 was higher in the NI forms in a significant way (*p*-value = 0.000). Higher ferritin levels were seen in NI forms at T1 and T2, with a significative correlation at T2 (*p*-value = 0.009). GGT showed a correlation with NI etiology at T1 and T2 (*p* = 0.024 and 0.038, respectively), instead of higher sodium levels which correlated with I forms at T1 (*p*-value = 0.004). The other laboratory tests performed at T1 and T2 did not correlate in a statistically significant way with the etiology of the fever. The results are illustrated in Table 5 and Table 6.

A correlation was found between the fecal calprotectin and the NI etiology (*p*-value = 0.008). The association between a positive blood culture and type I etiology was also significant (*p*-value = 0.008). However, there was no correlation between urinalysis, urine culture, throat-tonsillar swab culture, and the etiology of fever. ANA and anti-cardiolipin antibodies showed a significant association with the NI etiology (*p*-value = 0.000 and 0.003, respectively). A statistical significance was also found between previous EBV infection and type I etiology (*p*-value = 0.033). Previous SARS-CoV-2 infection seemed to correlate with NI fever (*p*-value = 0.002).

Regarding cardiological evaluation, echocardiogram showed anomalies in type I group, instead of ECG anomalies which were linked to NI forms. The alterations on abdominal ultrasound were associated with type I etiologies (*p*-value = 0.001). Conversely, pathological findings on chest and/or abdominal X-ray did not correlate with the nature of fever.

## 5. Discussion

The etiology of fever was predominantly infective in our study, followed by autoimmune/rheumatological forms (about 20% of cases), as also seen in other studies [17,18,19]. The pathogens of type I forms were mostly represented by viruses, in opposition to the literature data in which bacterial infections were the most common [20]. In addition, our study showed a significant association between higher mean age at the time of admission, family history for periodic fever, higher mean days with fever, and longer hospitalization in NI forms. These data were confirmed by other studies [18].

Regarding the frequency of admission for fever, the international literature shows a reduction in emergency room visits during the pandemic years. The frequency of community acquired infections was lower than the pre-pandemic period [20,21,22]. However, in the same years, it was highlighted that there was an increase in mononucleosis and urinary tract infections and appendicitis [23]. This could be a justification for the poor influence of the pandemic period on the hospitalization data. Parents’ fear of a possible SARS-CoV-2 infection and poorness of information about risks and severity caused a higher number of visits to the emergency room for fever with lower mean days of onset than the pre-pandemic era [23]. On the other hand, the fear of contracting the infection caused an increase in diagnostic delays [24,25,26]. The result was confirmed by our study where an increase in hospitalization for type I fever was seen over 2020 [23]. After the reopening of schools, the epidemiology of fever remained almost stationary, and new pathological entities emerged, such as COVID-19-related multisystem inflammatory syndrome [27].

The incidence of type I forms usually increases in the winter months, due to the permanence in closed and crowded places (such as schools and nursery schools) with reduced air exchange. The result is a greater circulation of pathogenic germs [28]. However, in our case series, we observed a peak incidence in August (26 cases equal to 11.8%), probably linked to febrile gastroenteritis and NI causes. On the other hand, the comparative analysis of the data did not highlight any correlation between the month of hospitalization and the etiology of the fever.

Regarding the antipyretic therapy, our study confirmed national and international indications on the use of paracetamol as a safe and effective molecule in the pediatric age [29,30]. Although guidelines advise against the combination of paracetamol and ibuprofen [29,30], we administered alternately/combinedly in 16.4% of cases, without recording any noteworthy adverse effects. Regarding the use of antibiotic therapy, our study confirmed the strong association between type I forms and the use of antibiotics. As can be seen, 79.5% of patients were treated against 71.8% with type I forms confirmed. This corresponds to the international guidelines, which suggest their use in cases of high suspicion for type I fever forms to avoid antibiotic resistance [20,31]. Another strong correlation was seen between NI forms and the infusion of intravenous immunoglobulins and systemic steroid therapy. This demonstrated that a good diagnostic suspicion allows for a targeted therapy.

The use of laboratory parameters is important for understanding the etiology of fever. In our population, the blood count showed high WBC and neutrophil levels with low Hb at T1. During fever, the inflammation usually causes a reduction of Hb values [32]. In some studies, anemia is strongly associated with mild type I form [33]. However, in our data, low Hb levels seemed to be correlated to NI forms of fever. Furthermore, a high lymphocyte’ count was strongly associated with type I fever forms, probably due to high incidence of viral forms in our population. In addition to anemia, a high PLT count was usually associated with NI fever forms, especially in cases of rheumatological diseases. This association was also seen in the international literature [27,34]. Regarding other laboratory parameters, CRP, ESR, and procalcitonin were high at T1, followed by a reduction until the discharge. As a demonstration of a progressive disease resolution, CRP and procalcitonin usually correlate with severe infective disease [35,36]. In our population, procalcitonin and ESR were not linked to the etiology of fever. This could be explained with the dosage performed in cases of high suspicion of severe infective disease. Conversely, CRP was highly associated with NI forms at T1. Another inflammatory marker is ferritin, an intracellular protein, which stores and releases iron according to body requests [37]. In this study, high ferritin levels at T2 were related to NI forms of fever, also seen in other studies concerning rheumatologic or hematologic diseases [27,34,38]. Finally, iron blood levels usually are reduced during infective fevers [39].

Recent research demonstrated that iron could be a prognostic factor for severe infective disease, such as SARS-CoV-2 [40]. In this study, iron levels were low at T1, and they did not link to type I fever. Other important biohumoral parameters are GOT and LDH, which could be elevated during fever or for blood sample collection. High GOT levels are usually associated with hepatic bacterial or viral involvement or in other rheumatological disease, such as atypical Kawasaki [41,42,43,44]. Contrariwise, LDH increases during tissue damage for blood sample collection, neoplasia, pneumonia, and more [45,46,47]. However, we did not find any correlation between the elevation of these parameters and the etiology of fever. On the other hand, high GGT levels seemed to be associated with NI fever during hospitalization, with a progressive increase from T1 to T2. This could be explained with liver involvement during treatments. Finally, low sodium levels at T1 in NI patients could suggest a possible prognostic marker for autoimmune diseases such as Kawasaki disease or COVID-19-related multisystem inflammatory syndrome [27,34].

According to clinical evaluation, we sometimes performed other tests to exclude disease like hemophagocytic lymphohistiocytosis (HLH). For this reason, we dosed triglycerides and fibrinogen levels [48]. In this research, high levels of fibrinogen at T1 and high levels of triglycerides at T2 did not show any correlation with the etiology of fever. For COVID-19-related multisystem inflammatory syndrome suspicion, we studied cardiac function and thromboembolic panel with D-dimer, T troponin, myoglobin, and pro-BNP [27]. In this study, these variables were elevated, probably due to cardiac involvement in rheumatologic diseases. However, no statistical significance was found with fever etiology.

One hundred and fifty patients performed other tests such as blood cultures or urinalysis, because urinary tract infections represent one of the most frequent causes of fever under the age of 2 years [49]. Based on anamnestic-clinical laboratory evaluation, some instrumental tests were carried out. While pathological findings on chest X-ray did not correlate with the type of fever, and anomalies found on abdominal ultrasound are associated with infectious etiologies. The percentage of patients subjected to instrumental investigations in our series was much lower than other studies, in which approximately 80% of patients underwent chest X-ray, and 47% underwent an ultrasound examination [17,19]. This evidence could be explained by our difficulty to perform instrumental tests for management and organizational problems. Regarding instrumental cardiological evaluation, a progressive attention has grown due to the advent of COVID-19-related multisystem inflammatory syndrome and its cardiac involvement [27]. Seventy-three echocardiograms and ECGs were performed, but the correlations with the etiology of the fever was not clear. According to our study, pathological alterations on the echocardiogram should be related to an infectious etiology, in opposition to other research where the autoimmune forms had important cardiac involvement [27,34]. On the other hand, electrocardiogram abnormalities appear to correlate with NI etiologies. Further studies are likely needed to consider cardiac involvement as a potential predictive marker of the etiology of fever.

Since the spring of 2020, following the definition of COVID-19-related multisystem inflammatory syndrome [27], serology for SARS-CoV-2 has been performed in hospitalized febrile patients to evaluate a previous SARS-CoV-2 infection. Eighty-two (37.3%) of our patients underwent this serology; 13 (15.9%) were found to have a previous infection or to have been vaccinated. This finding correlates significantly with the NI etiology of the fever. The other viral and bacterial serologies performed did not show any statistical correlations with the etiology of the fever. The most performed serology was EBV (52.7%), followed by CMV (44.1%) and adenovirus (40%). Fecal calprotectin, useful for identifying IBD at the onset [50], was performed in only six patients with an evident NI form. The literature supports this data. Not only IBDs are associated with high fecal calprotectin levels, but also other NI fevers such as the familial Mediterranean fever show a similar trend [51]. Finally, the autoimmunity assay was reserved for patients in whom there was a high clinical suspicion of autoimmune disease; ANA and anti-cardiolipin positivity was correlated to NI fever.

Despite the importance of the frequency of fever as a cause of medical assistance [2], the limitation of our study was the small number of the analyzed population. This was probably due to the coexistence of other departments like pediatric nephrology, in which patients with fever due to urinary tract infection (approximately 5% of causes of fever in the first 2 years of life [49]) were hospitalized. In addition, our center was not identified as a COVID-19 center during the pandemic period. For this reason, patients with SARS-CoV-2 infection were transferred to other pediatric hospitals. Eligibility and exclusion criteria further restricted the number of patients.

## 6. Conclusions

In line with the literature already present, in this study, childhood fever was usually infective, but an etiological agent was not always identified. Although the SARS-CoV-2 pandemic has affected hospital management and the social life of children, this has not led to significant changes in the number hospitalized patients for fever.

As is already known in the literature, high lymphocyte levels are correlated with type I fever, mostly viral. Regarding NI fever, new potential predictive factors that emerged from our analysis are as follows: older age at the time of hospitalization, family history of periodic fever, fever lasting more than 8 days, and longer hospitalization. In addition, laboratory variables that could guide us towards NI forms are represented by unexplained anemia, high PLT count during hospitalization, high CRP and ferritin, and hyponatremia at the time of admission. Further studies are needed to confirm these elements as predictive of NI etiology.

Finally, our data confirm the use of paracetamol as the preferred antipyretic treatment by pediatricians, due to its ease of handling and effectiveness.

## Data Availability

The original contributions presented in the study are included in the article, further inquiries can be directed to the corresponding author.

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
