# Peer review of "Retrospective Analysis of Fever in Pediatric Age: Our Experience over the Last 5 Years"

_children, 2024, doi:10.3390/children11050539_

Round 1

Reviewer 1 Report

Comments and Suggestions for Authors

The presented manuscript describes the results of a retrospective study with unclear aim to test markers of noninfectious versus infectious origin of fever in children. As explained in the manuscript, fever is one of the greatest cause of parental anxiety and the most common reason to visit pediatric emergency department, but also a symptom common to the widest possible range of various diseases. Therefore, the workup of fever requires a stepwise approach, especially since many rheumatological and other conditions requires exclusion of other etiologies in order to make the diagnosis. Unfortunately, raises many major concerns, such as the lack of the clear aim of the study, details on the criteria for hospitalization, the list of rheumatic and other diseases diagnosed in patients, diagnostic criteria for rheumatic and other diseases etc. Therefore, I suggest to address all of this issues, to add the algorithm on how patients were handled and to add the table with final diagnosis. 

Comments on the Quality of English Language

The whole manuscript necessitates a thorough editing by an English native speaker since it contains numerous grammatical and semantical errors. 

Author Response

Dear Reviewer,

We appreciate your suggestions for improving my manuscript.

The aim of the study is explained from line 77 to line 78.

The hospitalization criteria are explained from line 81 to line 83.

The list of rheumatic and other diseases are explained from line 208 to line 2010 and listed from line 215 to line 222. The table with all the diagnosis is present at page 6 of 14.

Criteria for non-infective disease are described from line 112 to line 114

Best regards

Reviewer 2 Report

Comments and Suggestions for Authors

The abstract needs complete rewriting as it does not explain the problem of the study. Please refer to the instructions for authors. 

I have major concerns about the study design and the observed results, which are in line with the study hypothesis. If you look at the SD for the major parameters like WBCs,PLT, ferritin, D-dimer, pro-BNP, and myoglobin in Tables 1, 2, 3, and 4, it is even more and sometimes more than the observed value (table 1 for ferritin and D-dimer, etc.).  As per my opinion, it is due to a difference in the fever etiology. In the case of infectious etiology, the values will be higher; on the other hand, non-infectious etiology values for these parameters will be lower, and the same thing happened here and resulted in a high standard deviation. 

I would highly request that authors segregate these values based on etiology and patients and put out manuscripts with different perspectives.

The methodology needs to be sectioned.

Also, include inclusion and exclusion criteria. 

The discussion is too lengthy; please summarize it for a better understanding. 

Comments on the Quality of English Language

Please check thoroughly for redundancy and repeatation. 

Author Response

Dear Reviewer,

We appreciate your suggestions for improving our manuscript.

The methodology is divided into a materials and methods section and a statistical analysis.

The inclusion and exclusion criteria are well described from line 81 to line 87

The discussion section is soo long in order to explain the results of our research. 

Best regards

Reviewer 3 Report

Comments and Suggestions for Authors

 This is a very interesting article regarding the to analyze the demographic and laboratory characteristics of patient hospitalized in A pediatric Center.

I think you have done a great job. The only comments I have to make are:

Line 47: remove “by”

Line 57: Based on the height -à Based on the elevation

Line 78: remove /

Line 104-105: rephrase as follows: “Due to the small number of some subgroups two large sets were considered for statistical purposes”

Line 121: Put “were” instead of “are”

Lines 132-136: Rephrase this sentence “Regarding the antipyretic therapy during the hospital stay, this is characterized by: 133 124 patients (56%) took paracetamol, 21 patients (10%) ibuprofen, 36 patients (16%) a com- 134 bination paracetamol-ibuprofen or alternating and 39 patients (18 %) have not taken any 135 antipyretic therapy” as follows: “Regarding the antipyretic therapy during the hospital stay, this was characterized by: paracetamol (133 124 patients - 56%), ibuprofen (21 patients – 10%), 36 patients (16%) a combination paracetamol-ibuprofen (36 patients – 16%) or alternating. Thirty- nine patients (18 %) have not taken any antipyretic therapy”

Lines 151 and 154: “(222,3 ng/ml ± 144,9)” and “233,4”-> remove , and put “.”

Libe 185: “Were performed” instead of “was performed”

Line 194: please put “73 patient performed abdominal …”

Comments on the Quality of English Language

Good

Author Response

Dear Reviewer,

We appreciate your suggestions for improving our  manuscript.

All the corrections suggested were performed.

Best regards

Round 2

Reviewer 1 Report

Comments and Suggestions for Authors

Unfortunately, in my opinion the aim is still not clear and the manuscript was not sufficiently edited to consider it for publication. 

Comments on the Quality of English Language

The manuscript still requires a substantial editing.

Author Response

Dear Reviewer, 
We appreciate you  for your precious time in reviewing our paper and
providing valuable comments.

It was your valuable and insightful comments that led to 
possible improvements in the current version.

The authors have carefully considered the 
comments and tried our best to address every one of them. We hope the manuscript after careful revisions meet your high standards.  All modifications in the manuscript have been highlighted in yellow.

Reviewer 2 Report

Comments and Suggestions for Authors

Please highlight all the changes you did in last revision. 

Author Response

Dear Editor,
We appreciate you for your precious time in reviewing our paper and
providing valuable comments. It was your valuable and insightful comments that led to possible improvements in the current version. The authors have carefully considered the comments and tried our best to address every one of them. We hope the manuscript after careful revisions meet your high standards. 
All modifications in the manuscript have 
been highlighted in yellow

Sincerely yours 

Prof. Mariella Valenzise 

Round 3

Reviewer 1 Report

Comments and Suggestions for Authors

I stick with my previous comments which have not been successfully addressed. Moreover, point by point response is lacking.

Comments on the Quality of English Language

Substantial English editing is required. 

Author Response

Dear Reviewer,

We apologie for the lack of a point by point response.

The aim of the study is better expressed from line 12 to line 15 and from line 22 to line 29 and also from line 77 to line 80.

Hospitalization criteria are present ( from line 83 to line 85).

Rheumatic and other diseases are listed in the table 3.

The diagnostic criteria for rheumatic and other diseases  are reported (from line 114 to line 120 + Scheme number 1).

The English has been  totally revised.

 All modifications in the manuscript have been highlighted in yellow.

We sincerly appreciate ypur suggestions which have given us the opportunity to  to improve our manuscript.

Sincerely yours 

Prof. Mariella Valenzise 

Reviewer 2 Report

Comments and Suggestions for Authors

Thank you for adding the necessary changes.

Author Response

Dear Reviewer , 

we are grateful for yours suggestions which have given us the opportunity to improve our paper 

Sincerely yours 

Prof. Mariella VALENZISE